# Quantifying Learnability and Describability of Visual Concepts Emerging in Representation Learning

**Iro Laina**     **Ruth C. Fong**     **Andrea Vedaldi**

Visual Geometry Group
University of Oxford
{iro, ruthfong, vedaldi}@robots.ox.ac.uk

## Abstract

The increasing impact of black box models, and particularly of unsupervised ones, comes with an increasing interest in tools to understand and interpret them. In this paper, we consider in particular how to characterise visual groupings discovered automatically by deep neural networks, starting with state-of-the-art clustering methods. In some cases, clusters readily correspond to an existing labelled dataset. However, often they do not, yet they still maintain an "intuitive interpretability". We introduce two concepts, visual learnability and describability, that can be used to *quantify* the interpretability of arbitrary image groupings, including unsupervised ones. The idea is to measure (1) how well humans can learn to reproduce a grouping by measuring their ability to generalise from a small set of visual examples (learnability) and (2) whether the set of visual examples can be replaced by a succinct, textual description (describability). By assessing human annotators as classifiers, we remove the subjective quality of existing evaluation metrics. For better scalability, we finally propose a class-level captioning system to generate descriptions for visual groupings automatically and compare it to human annotators using the describability metric.

## 1    Introduction

Recent advances in unsupervised and self-supervised learning have shown that it is possible to learn data representations that are competitive with, and sometimes even superior to, the ones obtained via supervised learning [27, 48]. However, this does not make unsupervised learning a solved problem; unsupervised representations often need to be combined with labelled datasets before they can perform useful data analysis tasks, such as image classification. Such labels induce the semantic categories necessary to provide an *interpretation* of data that makes sense to a human. Thus, it remains unclear whether unsupervised representations develop a human-like understanding of complex data in their own right.

In this paper, we consider the problem of assessing to what extent abstract, human-interpretable concepts can be discovered by unsupervised learning techniques. While this problem has been looked at before, we wish to cast it in a more principled and general manner than previously done. We start from a simple definition of a *class* as a subset $\mathcal{X}_c \subset \mathcal{X}$ of patterns (*e.g.* images). While our method is agnostic to the class generation mechanism, we are particularly interested in classes that are obtained from an unsupervised learning algorithm. We then wish to answer three questions: (1) whether a given class is interpretable and coherent, meaning that it can be indeed understood by humans, (2) if so, whether it is also describable, *i.e.* it is possible to distill the concept(s) that the class represents into a compact sentence in natural language, and (3) if such a summary description can be produced automatically by an algorithm.

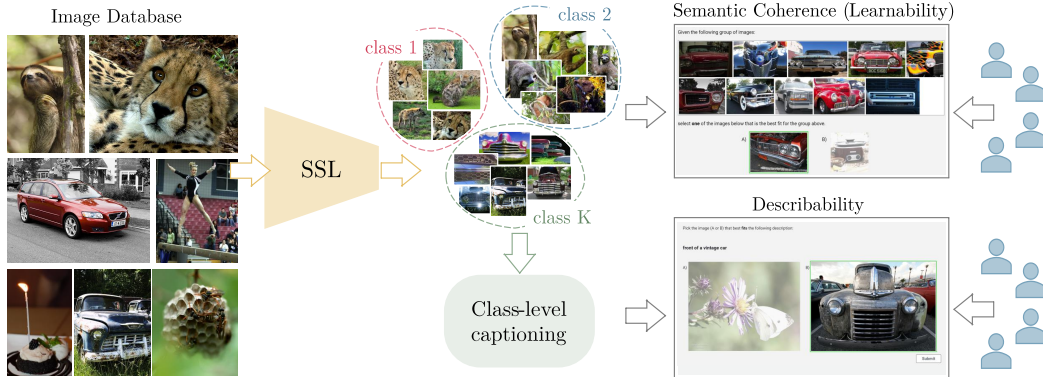

Figure 1: **Our framework**. We evaluate classes obtained by self-supervised learning algorithms (SSL) using human judgements. We formulate our evaluation as two forced-choice tasks and measure (a) the learnability of a class by humans and (b) the describability, *i.e.* the ability to distill the gist of the class into a description in natural language (class-level captioning).

The first problem has already been explored in the literature (*e.g.* [75]), usually using human judgment to assess class interpretability. In short, human annotators are shown example patterns from the class and they are asked to name or describe it. Unfortunately, such a metric is rather subjective, even after averaging responses by several annotators. In our work, we aim to minimize subjectivity in evaluating class interpretability. While still involving humans in the assessment, we cast the problem as the one of *learning* the class from a number of provided examples. Rather than asking annotators to identify the class, we test their ability to discriminate further examples of the class from non-class examples. The accuracy from this classification task can then be used as an objective measure of human learnability of the class, which we call *semantic coherence*. As we show later, self-discovered classes are often found to be semantically coherent according to this criterion, even when their consistency with respect to an existing label set — such as ImageNet [17] — is low.

Note that our semantic coherence metric does not require naming or otherwise distilling the concept(s) captured by a class. Thus, we also look at the problem of *describing* the class using natural language. We start by manually collecting names or short descriptions for a number of self-labelled classes. Then, we modify the previous experiment to test whether annotators can correctly recognise examples of the class from negative ones based on the provided description. In this manner, we can *quantify* the quality of the description, which is directly related to how easily describable the underlying class is.

Finally, we ask whether the process of textual distillation can be automated; this problem is related to image captioning, but with some important differences. First, the description has a well-defined goal: to teach humans about the class, which is measured by their ability to classify patterns based on the description. This approach also allows a direct and quantitative comparison between manual and automatic descriptions. Second, the text must summarise an entire class, *i.e.* a collection of several patterns, rather than a single image. To this end, we investigate ways of converting existing, single-image captioning systems to class-level captioning. Lastly, we propose an automated metric to validate descriptions and compare such systems before using the human-based metric.

## 2   Related work

**Unsupervised representation learning and self-labelling.**   In this work, we primarily study representations learned from unlabelled data. A number of methods use domain-specific, pretext tasks to learn rich features [18, 35, 52, 57]. Contrastive learning is another promising direction that has recently lead to several state-of-the-art methods [4, 13, 27–29, 55, 63, 65, 68], closing the gap to supervised pretraining. This paradigm encourages similar examples to be close together in feature space and non-similar ones to be far apart. A third research direction combines representation learning and clustering, *i.e.* jointly learning data representations and labels, in a variety of ways [3, 6, 8–10, 31, 33, 53, 70–72]. The representational quality of self-supervised methods is most commonly evaluated by performance on downstream tasks (*e.g.* ImageNet [17]), but relatively little work has been done on characterizing the kinds of visual concepts learned by them.

**Model interpretability.** A large body of work has focused on understanding the feature representations of CNNs; these typically focus on supervised networks. One approach is to *visualise* feature space dimensions (*e.g.* a single filter or a linear combination of filters). This can be done using real examples, *i.e.* by showing the top image patches that most activate along a given direction [74, 75], or by using generated examples (a.k.a. activation maximization or feature visualization) [46, 50, 51, 54, 62, 66]. Another approach is to *label* dimensions; this can be done automatically by correlating activations with an annotated dataset of semantic concepts [5, 19, 38, 77]. In particular, [5, 19] use this paradigm to compare supervised representations with self-supervised ones. This can also be done by asking human annotators to label or name examples via crowd-sourcing platforms like Amazon Mechanical Turk [24, 75]. Annotators can be used to compare the interpretability of different visualizations by asking them to choose the visualization they prefer [77]. Our work is most similar to [24, 75], yet we differ in that we replace the subjective nature of asking for free-form labels with an objective task that the annotators are asked to perform.

**Learning visual concepts.** Our approach to understanding the visual concepts learned by representation learning algorithms is to test whether they are "learnable" by humans. The work by Jia et al. [34] is similar in that they ask humans whether query images belong to a reference set of a visual concept. The human judgements then serve as the ground-truth signal to a learning system. In contrast, we use human judgements to assess the learnability of *machine*-discovered concepts. Relatedly, [78] measure the relatedness of human judgements and machine predictions in the context of adversarial images.

**Related topics in cognitive science.** Several works in cognitive science have also studied the learnability of concepts by humans, starting from word learning: Xu and Tenenbaum [69] let human subjects observe visual examples of a novel word and estimate the probability of a new example being identified as the same word/concept. This problem is closely related to the notion of representativeness [64], which is addressed in [1, 25] using natural image datasets. Work on representativeness, *i.e.* estimating how well an image represents a set of images, aims to model human beliefs with Bayesian frameworks. In the context of behavioral studies, [30] further attempt to couple Monte Carlo Markov Chains (MCMC) with people, through a series of forced choice tasks of selecting an image that best fits a description. The oddball paradigm [7], which asks subjects to pick the "odd" example out of a set of examples, is also related to our work.

**Image captioning.** Finally, we study the degree to which self-discovered concepts can be *described* automatically. This is similar to image captioning, except that the goal is to describe categories collectively, which requires reasoning about intra- and inter-class variation. While some methods focus on discriminability [12, 16, 43, 44, 67], this is only done on image level, with the goal of generating more diverse captions. Others aim to explicitly describe the differences between pairs of examples (*e.g.* video frames) [21, 23, 32, 56]. Most similar to our work is [41], which generates a short description for a target *group* of images in contrast to a reference group. Also related is the concept of visual denotation in [73], *i.e.* the set of images that depict a single linguistic expression.

## 3   Coherence measures

In this section, we introduce our measure of semantic coherence and describability for visual classes. Let $\mathcal{X}$ be a space of patterns, *e.g.* a collection of natural images, and $\mathcal{X}_c \subset \mathcal{X}$ be a given class. We construct $\mathcal{X}_c$ as follows: Given a learned, binary function $\phi_c(x) \in \{0, 1\}$, we define $\mathcal{X}_c = \{x \in \mathcal{X} : \phi_c(x) = 1\}$ as the space of input images that "activate" $\phi_c$. For deep clustering methods [3, 8], which assign images to clusters, $\phi_c$ is an indicator of assignment to cluster $c$. For unsupervised representation learning methods [13, 27], $\phi_c$ can be constructed by clustering the learned representations. For an arbitrary neural network, one can define $\phi_c$ by thresholding activations of filter $c$ in a given layer, as done in [5, 75].

**Class learnability and semantic coherence.** Previous work in deep clustering [8] suggests that the learned clusters are not exclusively object categories but often exhibit more abstract concepts, patterns, or artistic effects, which cannot be captured by comparing self-supervised representations to annotated datasets (*e.g.* ImageNet). Consequently, we wish to assess semantic groupings independent from any a-priori data labelling.

We formulate this as testing whether a class $\mathcal{X}_c$ is *semantically coherent* by testing if the class can be easily *learned* by humans. For this purpose, we do not require the class to be easily describable via

natural language; instead, we show the annotators examples from the class and ask them to classify further images as belonging to the same class or not, measuring the classification error.

Formally, given the class $\mathcal{X}_c$, we conduct a number of human-based tests in order to assess its learnability or semantic coherence. Each test $T$ is a tuple $(\hat{\mathcal{X}}_c, x_0, x_1, z, h \mid \mathcal{X}, \mathcal{X}_c)$ where $\hat{\mathcal{X}}_c \subset \mathcal{X}_c$ is a random subset of images from class $c$ with a fixed cardinality (*i.e.* $|\hat{\mathcal{X}}_c| = M$). Then, $x_0 \in \mathcal{X}_c - \hat{\mathcal{X}}_c$ is a random sample of class $c$ not in the representative set $\hat{\mathcal{X}}_c$, and $x_1 \in \mathcal{X} - \mathcal{X}_c$ is a random sample that does not belong to class $c$ (*i.e.* background). Finally, $z \in \{0, 1\}$ is a sample from a uniform Bernoulli distribution and $h$ is a human annotator, selected at random from a pool.

The human annotator $h$ is presented with the sample $(\hat{\mathcal{X}}_c, x_z, x_{1-z})$ via a user interface and her/his goal is to predict the value of $z$: $\hat{z} = h(\hat{\mathcal{X}}_c, x_z, x_{1-z})$. Intuitively, the annotator is shown a number of images $\hat{\mathcal{X}}_c$ from the class as reference and two query images $x_0$ and $x_1$, one which belongs to the class and one which does not, in randomized order. The annotator's goal is to identify which of the two images belongs to the class. We then define the *coherence* of the class as

$$C(\mathcal{X}_c) = \mathbb{E}_T[z = h(\hat{\mathcal{X}}_c, x_z, x_{1-z})], \tag{1}$$

that is the average accuracy of annotators in solving such tasks correctly. We thus evaluate the average ability of annotators to learn the class from the provided examples.

Finally, we consider different functions for sampling negatives which correspond to exploring different aspects of the learned classes. Specifically, we consider two different functions. The first and simplest choice is to sample negatives uniformly at random from $\mathcal{X} - \mathcal{X}_c$. This approach explores the overall learnability of a class. The second is to provide the annotator with a binary choice between a positive and a *hard negative* query. The hard negative image can be sampled from another class $\mathcal{X}_{c^*}$, where

$$c^* = \arg\min_{c' \neq c} |\alpha(\mathcal{X}_c) - \alpha(\mathcal{X}_{c'})| \quad \text{and} \quad \alpha(\mathcal{X}) = \frac{1}{|\mathcal{X}|} \sum_{x \in \mathcal{X}} f(x) . \tag{2}$$

In the above equation, $f(x)$ is a function producing a feature representation of image $x$. In other words, we sample hard negatives only from the class that is "most similar" to the target one based on class centroids in the feature space induced by $f$. Intuitively, this approach tests whether there are sufficient fine-grained differences between classes to be learnable by humans.

The advantage of Eq. (1), when compared to alternatives presented in prior work [24, 75], is that it does not require the class to be easily describable in words. It also provides a simple, testable and robust manner to assess the visual consistency of any image collection, including self-supervised classes. Note that there are more variants of the problem above. First, the problem presented to the annotator can be modified in various ways, *e.g.* presenting more queries. In particular, it is possible to also show explicitly more than one "negative" example (in our case, they are shown exactly one, $x_1$, v.s. $M + 1$ positive examples). Second, the difficulty of the problem can be modulated by the ambient space, and hence background images, differently. If all images in $\mathcal{X}$ are similar (*e.g.* only dogs), then separating a particular class $\mathcal{X}_c$ (*e.g.* a dog breed) is significantly harder than if $\mathcal{X}$ is more generic (*e.g.* random Internet images). We leave the exploration of these extensions to future work.

**Class describability and description quality.** We are also interested in assessing whether self-supervised classes capture concepts that can be compressed into a natural language phrase that describes the gist of the class. Such concepts might be represented by higher-level semantics, such as object categories or scenes (*e.g. puppy lying on grass*) or actions (*e.g. racing*), but they can also refer to mid-level characteristics (*e.g. striped texture*).

We extend the idea of semantic coherence to also include a free-form, language-based description for the class. The assessment task is the same as above, but instead of seeing a reference set of examples from a given class, annotators are shown a short description of the class. Prior work characterizes the describability of a class by asking annotators whether they believe a given class is semantic; this is a subjective assessment. Instead, we measure the effectiveness of a description in characterizing a class by its ability to convey useful information, *i.e.* to "teach" a human annotator about a class; this is an objective assessment of the utility of a description.

Formally, the protocol mentioned above is modified by replacing $\mathcal{X}_c$ with a description $D_c$ of the class $\mathcal{X}_c$ in natural language. We evaluate the *describability* of the class as:

$$C(\mathcal{X}_c, D_c) = \mathbb{E}[z = h(D_c, x_z, x_{1-z})]. \tag{3}$$

Eq. (3) captures the effect of two factors. The first one is the semantic coherence of the underlying class $\mathcal{X}_c$. When class samples exhibit low coherence, the class can neither be understood by annotators nor described precisely and compactly. The second factor is the quality of the description $D_c$ itself. Namely, given a fixed class $\mathcal{X}_c$, we can use Eq. (3) to assess different descriptions $D_c$ based on their efficacy in covering the information required to characterize the class. As a baseline in this experiment, we consider human-generated descriptions, which helps decouple the two factors.

## 4 Automatic class-level captioning

Given the above protocol for assessing class describability with human subjects, we next consider the problem of generating descriptions for arbitrary image collections automatically. We hereby refer to this problem as *class*-level captioning, emphasising the difference from *image*-level captioning. In particular, the goal in class-level captioning is to accurately describe not just a single image, but the entire collection — or the most representative part of it — which requires distilling the commonalities of images that fall under the given collection. Moreover, to encourage discriminativeness across various classes, such descriptions must be as specific as possible. For example, the description "organism" may accurately describe a class; however, it likely can also be applied to more than one class, and is thus an inadequate description.

As there exist no training data for class-level captioning, our approach draws inspiration from unsupervised text summarisation techniques [47, 49]. As a first step, we use a pre-trained captioning model $g : \mathcal{X} \to \mathcal{S}$ to generate descriptions $s = g(x)$ for each image $x \in \mathcal{X}$ independently. We then find the most representative description for each class from $\mathcal{S}_c = \{g(x) \mid x \in \mathcal{X}_c\} \subset \mathcal{S}$:

$$D_c = \underset{s \in S_c}{\arg\min} \left( \frac{1}{|\mathcal{S}_c|} \sum_{s^+ \in \mathcal{S}_c \backslash \{s\}} d(s, s^+) - \frac{1}{|\mathcal{S} - \mathcal{S}_c|} \sum_{s^- \in \mathcal{S} \backslash \mathcal{S}_c} d(s, s^-) \right). \tag{4}$$

In the above optimization, $d(\cdot, \cdot)$ is a distance metric between pairs of captions. Intuitively, we choose a caption that is close to other captions for a given class while simultaneously being far away from captions of other classes. This is done by selecting the caption that maximizes the difference between the intra-class ($\mathcal{S}_c$) and inter-class ($\mathcal{S} \setminus \mathcal{S}_c$) average caption distance.

We note that any metric suitable for evaluating language tasks can be used, such as ROUGE [42], which is commonly used in text summarisation. However, in order to better account for semantic similarities present in the captions, we define our distance function as $d(s, s') = 1 - \frac{\psi(s)^T \psi(s')}{\|\psi(s)\| \, \|\psi(s')\|}$, where $\psi : \mathcal{S} \to \mathbb{R}^n$ is a function mapping a sentence to an $n$-dimensional embedding space. This allows for sentences that have common semantic properties to be represented by similar vectors. Then, $d$ computes the cosine distance between two sentences in embedding space. We can obtain $\psi(\cdot)$ from the captioning model itself, or, in the general case, we can use an off-the-shelf sentence encoder that captures semantic textual similarities [11, 39, 59].

In contrast to image captioning, we do not evaluate our automatic descriptions directly against human-provided descriptions, due to known limitations of evaluation metrics for this task [15, 37]. Instead, here we can evaluate both automatic and human-generated descriptions using our describability metric, which measures how effective a description is in teaching humans to classify images correctly.

## 5 Experiments

Our experiments are organized as follows. First, we examine the representations learned by two state-of-the-art approaches, namely SeLa [3] and MoCo [27], and use our learnability metric (Eq. (1)) to quantify the semantic coherence of their learned representations. We then repeat theses experiments by providing human-annotated, class-level descriptions to measure the respective describability. We further validate the approach against selected ImageNet categories, which are highly-semantic by construction and for which an obvious description (*i.e.* the object class name) is readily available. Finally, we evaluate the automatic class-level descriptions that we obtain from our method.

Table 1: Human assessment of semantic coherence for self-supervised methods [3, 27] and ImageNet. We evaluate the semantic coherence of self-supervised classes using random (R) and hard (H) negatives. Results grouped by purity range.

| Method | Purity range | Semantic Coherence (R) | | | Semantic Coherence (H) | | |
| --- | --- | --- | --- | --- | --- | --- | --- |
| | | Mean | 95% CI | IRR | Mean | 95% CI | IRR |
| SeLa [3] | (0.3, 0.4] | 71.8 | [68.0, 75.4] | 32.8 | 55.3 | [51.3, 59.4] | 7.4 |
| | (0.4, 0.5] | 94.2 | [92.0, 95.9] | 87.7 | 60.0 | [56.0, 63.9] | 14.9 |
| | (0.5, 0.6] | 97.2 | [95.5, 98.3] | 95.3 | 71.8 | [68.0, 75.4] | 31.4 |
| | (0.6, 0.7] | 99.7 | [98.8, 100.0] | 99.5 | 63.2 | [59.2, 67.0] | 22.9 |
| | (0.7, 0.8] | 98.0 | [96.5, 99.0] | 94.9 | 65.3 | [61.4, 69.1] | 25.9 |
| | (0.8, 0.9] | 99.8 | [99.1, 100.0] | 99.8 | 63.8 | [59.8, 67.7] | 22.3 |
| | (0.9, 1.0] | 98.8 | [97.6, 99.5] | 98.0 | 72.2 | [68.4, 75.7] | 36.6 |
| MoCo [27] | (0.3, 0.4] | 89.8 | [87.1, 92.1] | 79.6 | 56.8 | [52.8, 60.8] | 9.2 |
| | (0.4, 0.5] | 93.8 | [91.6, 95.6] | 86.3 | 63.2 | [59.2, 67.0] | 14.4 |
| | (0.5, 0.6] | 96.5 | [94.7, 97.8] | 93.0 | 62.5 | [58.5, 66.4] | 25.8 |
| | (0.6, 0.7] | 98.8 | [97.6, 99.5] | 96.7 | 64.7 | [60.7, 68.5] | 24.4 |
| | (0.7, 0.8] | 100.0 | [99.4, 100.0] | 100.0 | 63.5 | [59.5, 67.4] | 20.5 |
| | (0.8, 0.9] | 99.5 | [98.5, 99.9] | 98.0 | 64.3 | [60.4, 68.2] | 22.1 |
| | (0.9, 1.0] | 99.2 | [98.1, 99.7] | 98.9 | 74.8 | [71.2, 78.3] | 52.0 |
| ImageNet | - | 99.0 | [98.3, 99.5] | 98.0 | - | - | - |

## 5.1 Assessing unsupervised image clustering

**Collection of human judgments.** To conduct experiments with human participants, we use Amazon Mechanical Turk (AMT). For each method (SeLa, MoCo, ImageNet), we select a number of classes and create 20 Human Intelligence Tasks (HITs) for each class. Each HIT is answered by 3 participants. In the following, we use data from the training set of ImageNet [17]. For the semantic coherence experiments, each HIT consists of a reference set of 10 example images randomly sampled from the class and two query images. The participants are asked to select the query image that is a better fit for the reference set. For the describability HITs, we retain the same query images but replace the reference image set with the class description, which is either manual or automatic. For the describability experiments, we restrict the number of HITs that a participant can answer to 1 per day *per class* such that it is not possible to unintentionally learn the class from the answers. Overall, 12k different HITs were answered by a total of 25,829 participants.

**Evaluation metrics.** The coherence of a class $\mathcal{X}_c$, as defined by Eq. (1), is the probability that annotators can identify the correct class out of a binary choice. In order to provide a higher-level analysis of our results, we report $C(\mathcal{X}_c)$ averaged over certain groups of classes, described below. In addition, we compute the 95% confidence intervals (CI) using the Clopper-Pearson method [14]. We also report the inter-rater reliability (IRR) using Krippendorff's $\alpha$-coefficient [40], where $\alpha = 1$ indicates perfect agreement, while $\alpha = 0$ indicates disagreement. Thus, low inter-rater agreement suggests that, in the forced-choice test given to the participants, the correct answer is not obvious.

**Semantic coherence on ImageNet classes.** In order to validate our methodology, we first conduct experiments on ImageNet [17], for which manually labelled categories exist, yielding $\mathcal{X}_c^{\text{IN}}$, $c \in \{1, \ldots, K\}$ with $K = 1,000$. As a sanity check, we report the average coherence over 20 selected ImageNet classesin the last row of Tab. 1. ImageNet labels are by definition highly semantic and consistent. Thus, as expected, the agreement of human participants with the ground truth labels is very high, reaching average semantic coherence of 99.0%.

**Semantic coherence on self-supervised classes.** Next, we evaluate two state-of-the-art, self-supervised learning methods. SeLa [3] simultaneously learns feature representations and predicts a clustering of the data by directly assigning images to clusters $\mathcal{X}_c^{\text{SeLa}}$ ($K = 3000$). We use the publicly available implementation based on ResNet-50 [26] and evaluate the clusters of the first out of 10 heads. In contrast, MoCo [27] does not produce its own labels, as it learns a continuous embedding space. To obtain $\mathcal{X}_c^{\text{MoCo}}$, we apply $k$-means on top of MoCo-v1 feature vectors (obtained using the official implementation) and set $k = 3000$ for a fair comparison with [3].

Table 2: Human assessment of describability for self-supervised methods [3, 27] and ImageNet. Class descriptions are either obtained manually (Human) or automatically (Auto). ImageNet descriptions are class names. Results grouped by purity range.

| Method | Purity range | Describability (Human) | | | Describability (Auto) | | |
| --- | --- | --- | --- | --- | --- | --- | --- |
| | | Mean | 95% CI | IRR | Mean | 95% CI | IRR |
| SeLa [3] | (0.3, 0.4] | 84.2 | [81.0, 87.0] | 63.7 | 78.7 | [75.2, 81.9] | 44.5 |
| | (0.4, 0.5] | 95.3 | [93.3, 96.9] | 90.1 | 93.3 | [91.0, 95.2] | 84.5 |
| | (0.5, 0.6] | 94.3 | [92.2, 96.0] | 87.2 | 94.5 | [92.4, 96.2] | 88.0 |
| | (0.6, 0.7] | 97.0 | [95.3, 98.2] | 96.0 | 96.0 | [94.1, 97.4] | 90.9 |
| | (0.7, 0.8] | 95.8 | [93.9, 97.3] | 91.8 | 94.8 | [92.7, 96.5] | 88.7 |
| | (0.8, 0.9] | 99.0 | [97.8, 99.6] | 98.2 | 98.2 | [96.7, 99.1] | 97.1 |
| | (0.9, 1.0] | 98.8 | [97.6, 99.5] | 98.4 | 97.3 | [95.7, 98.5] | 94.6 |
| MoCo [27] | (0.3, 0.4] | 78.3 | [74.8, 81.6] | 55.9 | 83.7 | [80.5, 86.5] | 60.4 |
| | (0.4, 0.5] | 95.0 | [92.9, 96.6] | 87.9 | 90.7 | [88.1, 92.9] | 76.4 |
| | (0.5, 0.6] | 96.5 | [94.7, 97.8] | 91.2 | 92.7 | [90.3, 94.6] | 84.7 |
| | (0.6, 0.7] | 96.3 | [94.5, 97.7] | 91.9 | 96.8 | [95.1, 98.1] | 94.8 |
| | (0.7, 0.8] | 98.0 | [96.5, 99.0] | 97.3 | 97.3 | [95.7, 98.5] | 95.0 |
| | (0.8, 0.9] | 97.8 | [96.3, 98.8] | 96.6 | 98.7 | [97.4, 99.4] | 98.2 |
| | (0.9, 1.0] | 98.5 | [97.2, 99.3] | 97.0 | 97.3 | [95.7, 98.5] | 95.9 |
| ImageNet | - | 95.6 | [94.3, 96.7] | 91.1 | - | - | - |

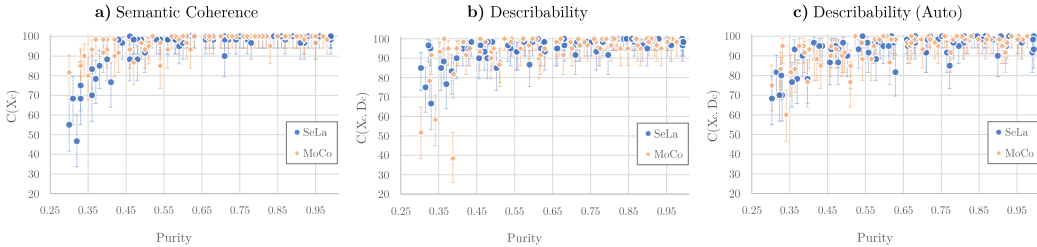

Figure 2: **ImageNet purity vs. semantic coherence and describability**. Each point represents a SeLa [3] or MoCo [27] class evaluated on AMT. Semantic coherence is shown for randomly sampled negatives (for hard negatives, please see the Appendix). Error bars correspond to 95% CI.

We note that there is no a-priori relationship between self-supervised classes (*i.e.* $\mathcal{X}_c^{\text{MoCo}}$ and $\mathcal{X}_c^{\text{SeLa}}$) and the human-annotated ones (*i.e.* $\mathcal{X}_c^{\text{IN}}$). We establish a relationship by computing the *purity* of a class $\mathcal{X}_c$ as $\Pi(\mathcal{X}_c) = 1 - \frac{H(l(\mathcal{X}_c))}{\log K}$, where $l : \mathcal{X} \to \{1, 2, \dots, 1000\}$ maps the contents of $\mathcal{X}$ to ImageNet labels and $H(l(\mathcal{X}_c))$ computes the entropy of the ImageNet label distribution within $\mathcal{X}_c$[1]. $\Pi(\mathcal{X}_c) = 1$ means that all the images in $\mathcal{X}_c$ share the same ImageNet label. Since in this case this label has been manually provided, high purity strongly correlates with high interpretability.

Tab. 1 reports the coherence for self-supervised classes over different purity ranges. Within each range, we sample 10 different classes for evaluation on AMT and report their average coherence. In the experiments using random negatives, high purity translates to high semantic coherence for both SeLa and MoCo, while very low purity generally corresponds to low coherence. We have observed that often such classes consist of bad quality images (*e.g.* blurry or grainy). Surprisingly, coherence shows a sharp increase with growing purity, also noticeable in Fig. 2. An interesting observation is that most of the classes of intermediate purity (0.5–0.8) appear to be highly coherent. This suggests that there exist self-supervised classes which are found to be "interpretable" yet do not align naturally with an ImageNet label. Some examples of such classes are shown in the Appendix.

On the other hand, using hard negatives we pose a stricter question, *i.e.* whether there are sufficient fine-grained differences between similar classes to make them learnable. It appears that this is often not the case, as suggested by the significant drop in coherence in Tab. 1. This is unsurprising given that the methods we analyse find a large number of clusters ($3\times$ the number of labels in ImageNet) and thus likely over-fragment the data. It also indicates that, while clusters are often semantically

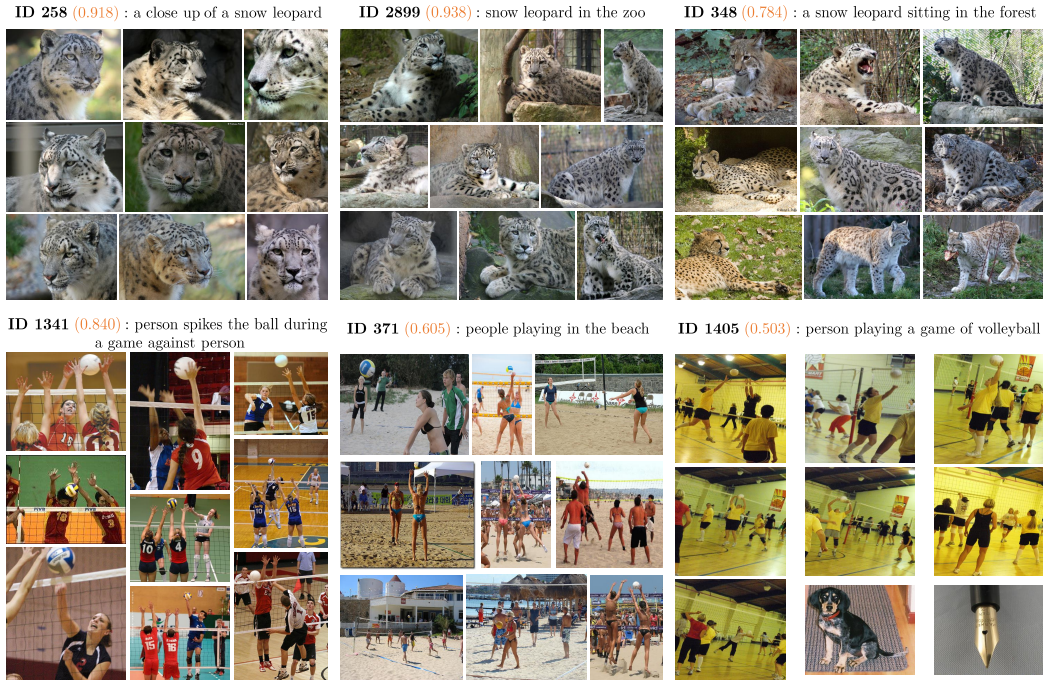

**ID 258** (0.918) : a close up of a snow leopard  **ID 2899** (0.938) : snow leopard in the zoo  **ID 348** (0.784) : a snow leopard sitting in the forest

**ID 1341** (0.840) : person spikes the ball during a game against person  **ID 371** (0.605) : people playing in the beach  **ID 1405** (0.503) : person playing a game of volleyball

Figure 3: Visualization of SeLa [3]-discovered classes with shared concepts (as defined by ImageNet labels); top: *snow leopard*; bottom: *volleyball*. Despite sharing a common concept, the cluster differences are easily recognisable by humans and are even reflected in the automatic descriptions.

coherent, they are not necessarily "complete", in the sense of encompassing all the images that should be grouped together. Finding the right number of clusters remains an open challenge in literature.

Although SeLa and MoCo are conceptually different methods, our assessment shows similar learnability scores for the resulting clusters, with MoCo showing a slight advantage in the lowest purity group with random negatives and in the highest for hard. To test whether the clusters themselves are similar, we compute their adjusted normalized mutual information (aNMI) against ImageNet, a standard metric to estimate similarity between clusterings. We obtain an aNMI of 40.7 for SeLa and 37.3 for MoCo. However, comparing SeLa to MoCo results in a higher aNMI of 44.0, which means the clusterings are more similar to each other than to ImageNet, despite their methodological differences. We also observe visual similarities qualitatively and present some examples in the Appendix.

**Describability.** Being coherent is not the same as being describable: the latter also requires that the "gist" of the class can be described succinctly in language. Next, we assess the describability of the above clusters, using manually written class descriptions. Learnability and describability are clearly correlated. While describability is generally slightly lower than coherence, sometimes we observe higher describability than coherence (Tab. 2). This occurs for classes where a description (*e.g.* "low quality photo") is more explicit and effective than a few visual examples in characterising the class. Still, lower coherence usually makes a class $\mathcal{X}_c$ harder to describe as a whole, and as a result the description might only capture a subset of $\mathcal{X}_c$.

## 5.2 Automatic class-level descriptions

**Implementation details.** To assess the describability of self-supervised classes with automatically generated descriptions, we first obtain captions for individual images using an off-the-shelf captioning model inspired by [60] (Att2in). We use a publicly available implementation[2] of the model trained on the Conceptual Captions dataset [61]. We then extract 1024-dimensional caption embeddings using Sentence BERT [59] (with `BERT-large` as the backbone)[3] trained on data from the Semantic

Table 3: Classification accuracy (%) of retrieved images (Google search). For binary accuracy we report the mean and standard deviation over 10 runs. We compare our approach to various baselines.

| | All classes | | | Selected classes | |
|---|---|---|---|---|---|
| Method | Top-1 | Top-5 | Binary | R@1 | R@5 |
| Random Image | 4.83 | 12.83 | $82.50 \pm 0.14$ | 7.16 | 17.62 |
| Representative Image | 4.05 | 17.41 | $86.00 \pm 0.15$ | 12.48 | 27.74 |
| ROUGE [42] | 6.12 | 16.06 | $87.77 \pm 0.09$ | 12.18 | 28.18 |
| $\mathcal{S}_c$ (w/o neg) | 7.64 | 19.10 | $89.48 \pm 0.07$ | 11.71 | 27.14 |
| $\mathcal{S}_c$ & $\mathcal{S} - \mathcal{S}_c$ | **7.88** | **20.29** | $\mathbf{91.18 \pm 0.07}$ | **12.57** | **30.21** |
| Manual description | - | - | - | 18.35 | 34.30 |

Textual Similarity benchmark [11], such that similar captions are represented by similar embeddings. Eq. (4) yields the most representative caption for the class, which we use as the class description.

**Evaluation.**  Our results on describability with automatic class descriptions are also shown in Tab. 2 and Fig. 2. Our findings show that the automatic system yields adequate descriptions for visual groups, with only a small gap to the respective manual/expert descriptions. In Fig. 3, we further show an interesting case of classes from [3], where each row shares a visual concept, according to ImageNet annotations. Nevertheless, differences between these classes are indeed observable, *e.g.* portrait photos vs. full-body views of the leopard or the difference in the background environment. We also notice the distinction between indoor and *beach* volleyball, which is not a label in ImageNet. Importantly, we note that these differences are also captured by the automatic class description, also shown in the figure. Further results are shown in the Appendix.

In addition, we propose an automated method to approximate describability and use it to compare our approach to various baselines. In Tab. 3, given a description for an unsupervised class, we retrieve $N = 10$ external images using a search engine (*e.g.* Google Image Search) and test whether these are classified in the same way by the unsupervised method (reporting Top-1 or Top-5 accuracy), thus weakly testing the quality of the description. We report this for all SeLa classes and for the ones used in the AMT experiments (computing recall: R@1, R@5). In order for this metric to be a proxy for our experiments with humans (*i.e.* binary tests), we also compute it for binary comparisons. For each image retrieved for $\mathcal{X}_c$ based on $D_c$, we randomly sample a negative among the images retrieved for other classes. We then compare their respective probabilities for class $c$ and report the percentage of correct outcomes, *i.e.* when the positive example is preferred.

Using this metric, in Tab. 3 we ablate our class-level descriptor generator and compare it to alternatives descriptions. First, dropping the negative term from Eq. (4)(w/o neg) results in a performance drop in all metrics; this indicates the importance of encouraging distinctive descriptions by contrasting against the other classes. Second, we compare to using ROUGE-L [42] as the distance function in Eq. (4), thus carrying out the optimization directly on word level. This results in a larger performance drop, highlighting the effectiveness of the sentence embedding space in extracting representative descriptions. Lastly, we compare to reverse image search, *i.e.* using an image as the search query. We experiment with both sampling a query image at random from a class and selecting the most representative image based on ResNet-50 features. In both cases, reverse image search yields significantly worse performance. Overall, we demonstrate our proposed method is a strong baseline on our describability assessments and also the closest to the results obtained using manual descriptions.

## 6   Conclusions

We have presented two novel concepts, visual learnability and describability, for quantifying the quality of learned image groupings. With rapidly improving performance of self-supervised methods, this quantification has three implications. First, understanding, describing and analyzing self-learned categories is an important direction in interpretability research. Second, when moving away from labelled data, it becomes necessary to evaluate methods without ground truth annotations. Finally, understanding the difference between human learnability and automatic class discovery can potentially lead to developing improved methods in the future.

## Broader Impact

**Interpretability tools for understanding feature representations.** Recently, a number of works have focused on explaining or interpreting deep learning models; such research is often known as explainable AI (XAI) or interpretability [22]. Due to the highly-parameterized nature of CNNs, most researchers treat such models as black-boxes and primarily evaluate them based on task performance on well-curated datasets (*e.g.* ImageNet classification). However, as deep learning is increasingly applied to high-impact, yet high-risk domains (*e.g.* autonomous driving and medical applications), there is a great need for tools that help us understand CNNs, so that we can in turn understand their limitations and biases. Our work contributes to the development of interpretability tools that can help society to responsible use and interrogate advanced technology built on deep learning. We primarily do this in two ways.

**Development of principled interpretability metrics.** First, we present a principled framework for evaluating the human-interpretability of CNN representations. While this may seem trivial, the interpretability research community has been lagging behind in the development of such metrics. There have been two main shortcomings of most interpretability evaluation: 1., they are often based on subjective or qualitative inspection, and 2., they fail to evaluate the faithfulness and interpretability of an explanation, that is, it should be both an accurate description of CNN behavior and easy-to-understand. These two shortcomings often go hand-in-hand. For example, [2, 20, 45, 58] highlight this issue for attribution heatmaps, which explain what parts of an image are responsible for the model's output decision. In particular, [2] shows that a number of attribution methods that are typically preferred for their visual appearance actually do not accurately describe the CNN being explained. Most metrics focus on evaluating the interpretability of an explanation without also measuring its faithfulness. This is a major limitation, as an explanation is not useful if it does not accurately describe the phenomenon being explained.

The typical methodology for human evaluation of CNN interpretability asks humans subjective questions like, "which explanatory visualization do you prefer or trust more?" [77], "do these images systematically describe a common visual concept?" [24], and "if so, name that concept" [76]. Such evaluations tend to evaluate the interpretability without faithfulness (*i.e.* how can we verify that this is the most accurate name for the concept?). In contrast, our work evaluates using both criteria by shifting from using humans as subjective annotators to using them as more learners that can be evaluated objectively. Our coherence metric objectively measures how interpretable a CNN-discovered cluster of images is, while our describability metric quantifies how faithfully a natural language description accurately characterizes such a cluster. We hope that our work serves as a springboard for future work that enables the use of human annotators in evaluating the interpretability of CNNs in a more principled manner.

**Understanding self-supervised representations.** Second, we focus on understanding self-supervised representations. Most work to date has focused on understanding CNNs trained for image classification.[4] However, supervised methods like image classifiers are limited in that they require expensive, manual annotation of highly-curated datasets. Thus, recent developments of self- and un-supervised methods is exciting, as they do not require manual labels. That said, there has been relatively little work dedicated to understanding self-supervised representations. The few works that do explore self-supervised representations typically apply techniques developed on supervised image classifiers to them [5, 19].

In contrast, we developed our evaluation paradigm with self-supervised methods in mind. In particular, we were motivated to develop an evaluation framework that could measure the interpretability of coherent, visual concepts that fall outside the limits of being described by labelled datasets. For example, in Fig. 3, we show that one self-supervised method discovered distinct clusters that highlight different environments of the same concept (*e.g.* different environments for playing volleyball). Standard interpretability methods of describing such clusters using a labelled dataset [5, 19] would likely map them onto the same label (*e.g.* "volleyball") and fail to characterize the subtle nuances captured by different clusters. Lastly, by design, our paradigm is agnostic to method and can also be used to understand other kinds of image representations, including non-CNN ones. We hope our work encourages further research on understanding other kinds of representations beyond image classifiers and developing interpretability methods explicitly for those settings.

## Acknowledgements and Funding Disclosure

We would like to thank Yuki Asano and Christian Rupprecht for helpful discussions and for their feedback on this work. We are also grateful for the EPSRC programme grant Seebibyte EP/M013774/1 (I.L.), ERC starting grant IDIU 638009 (I.L), and Open Philanthropy Project (R.F.).

## Footnotes

[1]In cluster analysis, entropy is used as an external measure of cluster quality [36].

[2] https://github.com/ruotianluo/GoogleConceptualCaptioning

[3] https://github.com/UKPLab/sentence-transformers

[4]This machine learning workshop highlights this over-emphasis and encourages more diverse XAI work.

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
