[Supplementary Material]

# Quantifying Learnability and Describability of Visual Concepts Emerging in Representation Learning

# Appendix

## 1 Qualitative Examples and Discussion

In Figure A3 and Figure A4 we show a selection of SeLa and MoCo classes respectively with varying purity (the lowest purity class for both methods has $\Pi \approx 0.3$). We can make several interesting observations.

First, we observe that concepts emerging in self-supervised methods, might not necessarily be annotated in ImageNet. A prominent example is shown in Figure A3(g), that is a cluster of *babies*, while the most frequent ImageNet label in this cluster is *bassinet*. Notably, while the cluster in question is of seemingly intermediate purity, it exhibits strong semantic coherence and describability, supporting our findings in Table 1 (main paper). It is therefore worth asking what is the implication of this on linear classification accuracy on ImageNet (*i.e.* training linear probes) as a way evaluate such methods; we leave this to future work.

Second, the quality of the class descriptions is often dependent on cluster quality. In most cases, *very* low purity clusters have no visual similarities and the concept shared among images (if any) is often difficult to identify, even for humans. For example, Figure A4(l) could be *"motion blur in dynamic scene"*. In these cases, the automatic class-level description is often unsuccessful in conveying the gist of the class (*e.g.* Figure A3(l), Figure A4(l)); this is also in agreement with low learnability and describability scores reported in Table 1 (main paper).

Third, we often observe clusters such as the ones shown in Figure A3(h), (j), where the prominent visual concepts could perhaps be identified as *"felt material"* and *"black circle"* respectively. In such cases, images in the class might be closer visually rather than semantically, *i.e.* sharing material or other visual properties. While these might be easily recognisable by humans, describing such concepts remains still a challenge for image captioning systems given that existing datasets are highly semantic — *i.e.* descriptions of textures, patterns, etc. are heavily under-represented. As a result, summarising the class by extracting abstract class-level descriptions from image-level semantics is difficult and sometimes results in failure cases. For the examples above, the automatic class-level descriptions "collapse" to the most prominent semantic concepts in each cluster, *i.e.* *"tennis ball"* and *"cup of coffee"*. Empirically, we have found that advancing class-level captioning systems will require additional knowledge about the world, for example how individual semantic categories can be linked on different levels, such as material (what are A and B made of?), appearance (how do A and B look like?) or usage (how are A and B used?).

In Figure A7 we show a few of clusters that we evaluated on AMT (20 HITs each), together with their learnability and describability scores. For each case, We also show samples from the hard negative cluster.

Table A1: Clustering quality metrics for all permutations of SeLa, MoCo and ImageNet.

| Method | NMI | aNMI | ARI |
|---|---|---|---|
| ImageNet–SeLa | 49.6 | 40.7 | 8.3 |
| ImageNet–MoCo [1k] | 43.4 | 39.5 | 6.0 |
| ImageNet–MoCo [3k] | 46.4 | 37.3 | 5.3 |
| SeLa–MoCo [1k] | 54.2 | 46.4 | 6.8 |
| SeLa–MoCo [3k] | 58.0 | 44.0 | 9.3 |

Figure A1: Semantic coherence measured with hard negatives (shown per cluster with 95% CI)

## 2 Additional Results

### 2.1 Learnability with Hard Negatives

In addition to Figure 2 from the main paper, we plot the learnability for each class with hard negative sampling in Figure A1. Because of over-fragmentation of the data into clusters, the "hard negative" counterpart is often a very similar (or even the same) concept, thus resulting in several pure clusters having low learnability scores, despite being highly interpretable. Some examples are shown in Figure A7 Learnability with hard negatives alone is not a good indicator of the overall coherence of a class.

### 2.2 Comparing Clusterings

In Table A1, we report the normalized mutual information (NMI), adjusted NMI and adjusted rand index (ARI) to quantify the quality of the clustering against ImageNet. We consider two independent clusterings for MoCo, one with 1k and one with 3k classes. Interestingly, as discussed in the main paper, we find out that there are more similarities between self-supervised methods SeLa and MoCo than similarities between each method and ImageNet, despite their being fundamentally different approaches. This holds true even for MoCo-1k, even though the number of clusters in this case is the same as with ImageNet labels.

We can also justify this observation qualitatively, as there exist clusters that are very similar to each other, yet they have intermediate to low purity with respect to ImageNet. Some interesting examples are shown side-by-side in Figure A5. This finding suggests that self-supervised approaches discover similar concepts — which are also *interpretable* by humans — and that these concepts might not necessarily align with ImageNet labels. Notably, in the examples shown in Figure A5, similar clusters also have a similar degree of "impurity" ($\Pi(\mathcal{X}_c)$ given in orange).

### 2.3 Evaluating Caption Quality

In addition to the describability experiments, described in the main paper, we also directly assess the quality of the automatic class-level captions by asking human participants to provide a rating. Specifically, given a set of images from a class[1] and the corresponding class caption, we ask workers to rate the effectiveness of the caption in describing the group *as a whole* using a Likert scale from 1 to 5 (1-Extremely bad, 2-Bad, 3-Adequate, 4-Good, 5-Excellent). We also ask whether the description is suitable for *at least one* image in the group, *i.e.* whether it is a partial description (answer: yes/no), which is particularly meaningful for impure classes. The results are shown in Figure A2. We observe that the outcome follows a trend similar to the one shown for describability in the main paper. We found that in some cases, class descriptions were rated low, even for highly coherent clusters, due to the inability of the captioning model to identify fine-grained categories in individual images.

Figure A2: **Evaluation of class-level caption quality. (a)** 2D histogram of human ratings for the quality of captions for a group as a whole, in a scale from 1 to 5. Each bin summarises all clusters over a purity range, e.g. 0.5–0.6. **(b)** We also ask whether the provided caption adequately describes at least one image in the group. The plot shows the average answer for clusters in each purity group, (1/yes, 0/no).

Table A2: Most common captions occurring in self-supervised classes.

| Rank | SeLa captions | Count | MoCo captions | Count |
|------|---------------|-------|---------------|-------|
| 1 | biological species perched on a branch | 27 | biological species perched on a branch | 36 |
| 2 | dogs playing in the grass | 21 | dogs playing in the grass | 18 |
| 3 | dogs sitting on the floor | 19 | dogs in the grass | 17 |
| 4 | a dog in a dog | 18 | a dog with a dog | 16 |
| 5 | a dog with a dog | 17 | biological species in the grass | 14 |
| 6 | biological species in the grass | 14 | a dog in a dog | 13 |
| 7 | dogs in a field | 11 | dogs sitting in the grass | 13 |
| 8 | dogs in the grass | 11 | a monkey sitting in a tree | 12 |
| 9 | a grasshopper on a leaf | 11 | a snake on the road | 11 |
| 10 | image may contain person playing a musical instrument on stage and indoor | 11 | a spider on a leaf | 11 |

## 2.4 Uniqueness of Captions

To assess the discriminative ability of the captions, we also report the number of unique captions over all 3000 classes, which is 2136 for SeLa and 2070 for MoCo. Ignoring stopwords, the number of unique captions becomes 1931 and 1859 respectively. We report the 10 most frequent class descriptions in Table A2. We then easily observe that these captions correspond to clusters consisting of fine-grained categories, such as breeds of dogs, birds or insects. In fact, while some of these clusters are relatively pure, most of them are not, suggesting that self-supervised algorithms in question cannot always learn such fine-grained distinctions. As an aside, conventional captioning models cannot exhaustively recognize fine-grained categories either. As an example, we show dog clusters in Figure A6, found by searching the class-level descriptions with queries "dog AND grass" and "dog AND sleeping". We verify that the self-supervised algorithms tend to group images by fur, color, environment, activity (*e.g.* playing with other dogs, sleeping, etc.) and even pose or viewpoint rather than distinguishing among dog breeds. We again observe similar behavior by both algorithms.

**(a) ID: 1908** (0.952)
a herd of zebra in the field

**(b) ID: 503** (0.930)
person runs with a ball during
a training session

**(c) ID: 160** (0.889)
mushrooms growing on the forest

**(d) ID: 2646** (0.869)
a white picket fence in front of a garden

**(e) ID: 2206** (0.717)
the arch in the old city

**(f) ID: 680** (0.680)
a lizard on a tree branch

**(g) ID: 393** (0.668)
a newborn baby lying on a bed

**(h) ID: 1330** (0.598)
tennis ball on a tennis ball

**(i) ID: 0** (0.558)
a shopping cart isolated on
white background

**(j) ID: 460** (0.455)
close up of a coffee cup

**(k) ID: 227** (0.419)
view from the window in the room

**(l) ID: 2568** (0.377)
a red door in the house

Figure A3: Samples from SeLa classes along with the predicted class captions, sorted by purity (in orange).

**(a) ID: 884** (0.986)
a monarch butterfly on a flower

**(b) ID: 2390** (0.907)
group of pink flamingos in the zoo

**(c) ID: 2129** (0.861)
a close up of corn on the cob

**(d) ID: 2489** (0.793)
person and his daughter on a carousel

**(e) ID: 1671** (0.709)
divers swimming in the sea

**(f) ID: 2888** (0.671)
a slice of bread in a baking dish

**(g) ID: 706** (0.610)
a skier on a snowmobile in the snow

**(h) ID: 1780** (0.587)
bottles in a bar

**(i) ID: 1027** (0.532)
a woman prepares food in
a restaurant

**(j) ID: 2335** (0.459)
view of the mountains from the lake

**(k) ID: 338** (0.437)
cup of coffee on black background

**(l) ID: 1264** (0.338)
a dog running on the water

Figure A4: Samples from MoCo classes along with the predicted class captions, sorted by purity (in orange).

SeLa

**ID: 1663** (0.807)

a spider on the web

**Top labels:**
spider web (210)
barn spider (117)
garden spider (51)
black and gold
garden spider (14)
lacewing (3)

**ID: 813** (0.681)

a white swan in the pond

**Top labels:**
pelican (88)
sulphur-crested
cuckatoo (62)
spoonbill (60)
american egret (47)
flamingo (40)

**ID: 379** (0.555)

a wooden box for the table

**Top labels:**
crate (122)
chest (57)
cradle (14)
plane (14)
rule (12)

**ID: 76** (0.456)

close up of a rusty metal on
a wooden table

**Top labels:**
hammer (35)
nail (26)
padlock (22)
tile_roof (20)
hatchet (16)

MoCo

**ID: 24** (0.836)

spider on a web

**Top labels:**
spider web (124)
barn spider (46)
garden spider (11)
black and gold
garden spider (6)
wolf spider (3)

**ID: 1487** (0.678)

biological species in a field

**Top labels:**
white stork (73)
spoonbill (47)
american egret (39)
crane (38)
albatross (29)

**ID: 30** (0.604)

a box of wood in a room

**Top labels:**
crate (111)
chest (31)
cradle (21)
file (19)
tub (13)

**ID: 2872** (0.498)

old rusty metal padlock on
a wooden door

**Top labels:**
padlock (146)
hammer (26)
iron (18)
chain (15)
hook (13)

Figure A5: Similar concepts emerging from different self-supervised methods, shown side by side. The top ImageNet labels in each cluster are shown on the right, with the number in parenthesis being the number of occurrences for each category. Notably, corresponding clusters also exhibit similar purity (in orange).

**Query: {dog, grass}**

SeLa

**Top labels:**
Chow (156)
Keeshond (41)
Leonberg (28)
Pomeranian (27)
Tibetan Mastiff (18)

**Top labels:**
Staffordshire
bullterrier (62)
Doberman (59)
Mexican hairless (27)
Rottweiler (20)
Kelpie (19)

MoCo

**Top labels:**
Bouvier des Fladres (70)
Curly-coated retriever (53)
Irish water Spaniel (44)
Kerry blue Terrier (33)
Giant Schnauzer (21)

**Top labels:**
Whippet (19)
Borzoi (15)
Beagle (14)
Malamute (12)
Pug (11)

**Query: {dog, sleeping}**

SeLa

**Top labels:**
Redbone (35)
Italian greyhound (28)
Whippet (24)
Basenji (23)
Basset (17)

**Top labels:**
Miniature Pinscher (55)
Rottweiler (31)
Appenzeller (25)
Staffordshire
bullterrier (23)
Doberman (20)

MoCo

**Top labels:**
Redbone (47)
American Staffordshire
terrier (38)
Weimaraner (36)
Bull Mastiff (36)
Italian greyhound (33)

**Top labels:**
Italian greyhound (23)
Weimaraner (20)
French bulldog (16)
Ibizan hound (14)
Redbone (13)

Figure A6: Examples of "dog" clusters found by SeLa and MoCo. We observe that *both* methods have learned to group images by color, fur, environment (*e.g.* grass, bed), etc. instead of fine-grained dog breeds.

| Evaluated Cluster | Hard Negative Cluster |
|---|---|

**SeLa, ID: 1068** (0.990)

**Manual Desc.:**
close up of coral fungi

**Automatic Desc.:**
close up of the flower

Learnability (R):  100.0%
Learnability (H):  78.3%

Describability (M):  98.3%
Describability (A):  91.7%

**SeLa, ID: 917** (0.974)

**SeLa, ID: 332** (0.542)

**Manual Desc.:**
close up of small snake or
lizard in human hand

**Automatic Desc.:**
a snake on a hand

Learnability (R):  98.3%
Learnability (H):  66.7%

Describability (M):  98.3%
Describability (A):  100.0%

**SeLa, ID: 2634** (0.597)

**MoCo, ID: 2310** (0.721)

**Manual Desc.:**
close up of amphibian
with big eyes

**Automatic Desc.:**
a frog on a leaf

Learnability (R):  100.0%
Learnability (H):  75.0%

Describability (M):  96.7%
Describability (A):  98.3%

**MoCo, ID: 2051** (0.536)

**MoCo, ID: 2201** (0.542)

**Manual Desc.:**
person working with their hands

**Automatic Desc.:**
hands of a potter making
a pottery wheel

Learnability (R):  85.0%
Learnability (H):  71.7%

Describability (M):  100.0%
Describability (A):  88.3%

**MoCo, ID: 1681** (0.400)

Figure A7: Examples of clusters that were evaluated. On the left, we show the target cluster. The middle column shows the manual (M) and automatic (A) descriptions obtained for the cluster and the evaluation outcome: learnability with random (R) and hard (H) negatives and describability. On the right, we show samples from the hard negative cluster used for the evaluation of learnability (H).

# 3   Evaluation Details

In the following, we provide the full list of classes used to collect human judgements on AMT, to quantify learnability and describability.

**Selected ImageNet categories.**   (387) lesser panda, (145) king penguin, (685) odometer, (321) admiral, (991) coral fungus, (916) website, (549) envelope, (76) tarantula, (807) solar dish, (103) platypus, (813) spatula, (731) plunger, (749) quill, (910) wooden spoon, (747) punching bag, (466) bullet train, (974) geyser, (640) manhole cover, (340) zebra, (323) monarch.

**Selected SeLa classes (sorted by descending purity).**   2777, 1296, 1537, 933, 1068, 1987, 527, 2434, 396, 2813, 1961, 977, 1332, 1139, 1802, 915, 115, 288, 2047, 136, 184, 1144, 85, 1476, 2375, 1761, 19, 222, 13, 624, 214, 813, 2296, 1993, 1278, 1042, 1406, 1285, 162, 2194, 2090, 1055, 420, 2857, 0, 379, 332, 500, 2250, 39, 63, 1026, 161, 2381, 8, 11, 76, 2233, 2523, 1246, 240, 2258, 338, 2867, 991, 796, 407, 1926, 327, 186.

**Selected MoCo classes [3k] (sorted by descending purity).**   1386, 252, 844, 1175, 785, 2823, 618, 658, 1501, 1597, 994, 76, 2109, 2960, 240, 924, 1451, 239, 442, 1799, 2881, 1725, 2030, 892, 1243, 370, 2416, 2310, 2102, 2673, 515, 1341, 2595, 2888, 158, 1165, 171, 2964, 608, 238, 350, 1470, 1026, 2612, 2497, 2201, 2992, 253, 169, 2311, 967, 66, 2074, 2003, 2335, 1865, 1730, 2195, 1510, 283, 311, 2490, 1221, 2128, 1206, 1711, 429, 1507, 1601, 410.

## Footnotes

[1]The clusters and image sets are the same as those used for measuring learnability (semantic coherence).