[Reviews · NeurIPS 2020]

Review 1

Summary and Contributions: This paper considers the problem of evaluating the quality of unsupervised image representations without using labels. The paper introduces two metrics based on human judgement: learnability (“Can a human determine if a new image belongs in a given group of images or not?”) and describability (“Can a human determine if an image is in a group or not based on a text description of the class?”). The paper also compares automatically generated captions to human-generated captions for evaluating describability. The proposed metrics are evaluated both quantitatively and qualitatively.

Strengths: The problem of evaluating unsupervised representations is a timely and important one, given the recent surging interest in unsupervised/self-supervised representation learning. In particular, the problem of assessing representation quality without imposing a set of labels is interesting and under-studied. The learnability and describability concepts seem like very good ideas - replacing free-form responses (“What do these images have in common?”) with an objective but still human-based metric (“How often is a human able to classify a new image correctly given a set of examples?”) strikes me as a simple but very useful perspective shift. The experimental evaluation is thorough, thoughtful, and multifaceted. Most reported results include a quantification of repeatability / stability (confidence intervals or other error bars), which is very helpful for understanding how the method works in practice.

Weaknesses: The paper only evaluates two unsupervised representation learning techniques: SeLa and MoCo. It is not clear why these two techniques were chosen in particular - were they the only two attempted? It would help to establish the generality of the proposed method if results were presented on a few more representation learning techniques. How were the qualitative results in Figure 3 selected? Are they meant to be average case results or best-case results? What do failure cases look like?

Correctness: The proposed methods seem reasonable.

Clarity: This paper is very well-written. The figures and tables are generally effective and clear.

Relation to Prior Work: The paper seems to be well contextualized.

Reproducibility: Yes

Additional Feedback: In L217 it is noted that the Mechanical Turk experiments are carried out “over a long period of time, to avoid biasing the participants” - what are the relevant details there? How long, and at what intervals? What is the motivation for the definition of “purity” given in L241? Is this a common metric (reference?) or is it something being proposed in this paper? What are the error bars in Figure 2? Are they also 95% CIs? Minor comments: There seems to be an incomplete sentence on L184. ***Post-Rebuttal Comments*** I think the rebuttal was satisfactory. After reading the rebuttal and the other reviews, I will maintain my previous rating in favor of acceptance.


Review 2

Summary and Contributions: The paper proposes a method to evaluate visual concepts learned in unsupervised settings based on two dimensions: 1) ability of a human to learn the concept with few examples, and 2) ability of a human to describe the concept in short natural language description. The computation of this metrics is done through crowd-sourcing through AMT. The paper also proposes a semi-automatic variant of the describability metric using an automatic captioning model. ** Post rebuttal** Authors addressed my concern regarding the hard negatives with new results. While I am not entirely convinced of the practical usefulness of task agnostic evaluation, I concede that it is an interesting problem which should be studied and the paper makes a good contribution towards this direction. The qualitative insights provided in the paper as pointed out in the rebuttal are also interesting. Considering these things, I will increase my rating to 7 and would be happy with the paper being accepted.

Strengths: Novelty:- Learnability and Describability is an intuitive and novel way to assess the quality of a representation/grouping. The proposed evaluation protocol tries to make the metrics less subjective by removing the need for humans to label clusters as done in prior work. The paper also proposes a solution to make part of the pipeline automatic, through a simple solution to generate cluster-level captions automatically. Significance and relevance :- Given the increased interest in self-supervised representation learning recently, the proposed metrics to evaluate these representation will be of interest to the community. The solution to make the evaluation metric automatic makes it more scalable and open to wider adaptation. The empirical evaluation and analysis is fairly good, barring the one criticism in the weakness section.

Weaknesses: I have two main criticisms of the work, one related to soundness of claims, second related to significance. Soundness:- All the proposed evaluation metrics are based on a paired binary classification task, where a human has to decide which of the two images (one positive other negative) belong to the shown class. The difficulty of the task and hence the measure of coherence of the cluster would depend a lot on this negative sample. However in the paper, the negative samples are picked randomly from outside the cluster. This probably makes the task much easier than if a related negative sample was used and could be the reason the methods score very high on relevance on table 1. This is also seen in the results in table 2. The network does much better on the binary classification score metric than the top-1/top-5 metrics, indicating the binary classification against a random sample is much easier. The paper very briefly mentions this in L146, but I think it should be investigated further, as it is central to usefulness of all the proposed metrics. Significance:- While the proposed metrics make sense intuitively, I am not clear how useful this evaluation is, given the associated costs of running human evaluation. The paper does not provide enough insights derived based on the metrics, or make a case for how this would be useful in practice. In what scenarios would we want to evaluate the representations without a downstream task ? Would the proposed metrics correlate better with some measure of usefulness of the learnt representation (eg. generalization to different domains ? or performance on some tasks?)? While this might be a bit hard, I think it would make the paper much stronger if there is an evaluation/discussion of usefulness.

Correctness: The empirical experiments are sufficient to validate the claims.

Clarity: The paper is very well written and easy to read.

Relation to Prior Work: Yes, to the best of my knowledge

Reproducibility: Yes

Additional Feedback: I have a few questions regarding the metrics and analysis. - How does size of a cluster affect its coherence and describability ? Intuitively, it seems smaller clusters would more likely be more coherent. Should the cluster size be compensated for in the evaluation? A related question would be, how do we compare different methods with differing amounts of clusters ? - Is there any relation between the length of the generated/human written description and the coherence of the grouping ?


Review 3

Summary and Contributions: The paper proposes a method to characterise visual groupings discovered by unsupervised/semi-supervised feature learning methods by introducing two concepts: visual learnability and describability that the authors aim to use for quantifying the interpretability of arbitrary image groupings. The learnability measure is defined as humans' ability to reproduce a grouping by generalising the discriminative class properties from a small set of visual examples. If the visual features are class specific, the humans should be able to group images from the same class together. The describability measure is defined as the classifier's ability to group images based on textual descriptions. If the description contains enough information about the class, then a feature extracted from the description should be classified correctly. The experiments are conducted using two popular unsupervised feature learning methods SeLa and MoCo on the large scale ImageNet dataset.

Strengths: Quantifying the interpretability of the learned representations of the unsupervised feature learning methods is an interesting research direction that is of high relevance to the explainable AI and representation learning communities. The distance based evaluation metrics presented in this submission are simple and easy to understand. As such, the reviewer assesses the reproducibility as high. The empirical evaluation is conducted on ImageNet with 1000 classes, one of the most challenging image classification datasets.

Weaknesses: The learnability measure requiring human evaluation limits the applicability of the metric to fine-grained scenarios where explanations are most desired, i.e. it is difficult to assess the grouping of the features when the features represent visually similar objects that are not familiar for a lay person. The describability measure has been presented in prior works (not cited) such as Generating Visual Explanations, Hendricks etal, ECCV 2016. The descriptions presented in Figure 3 raises several concerns: 1. The class label is included in the description, hence the description reveals the solution. Arguably 'a close up of a snow leopard' is not a description of a snow leopard but the description of the pose of the object. If the task is not classification, i.e. distinguishing snow leopards from black panthers, but rather retrieval, i.e. finding snow leopards taken from similar camera angles, this needs to be made clear in the paper. If the task is classification, the class label should not appear in the description but rather the description should focus on the discriminative visual properties of the object. 2. If the task is classification as indicated in the paper, then a natural testbed would be the already existing natural language based explanation datasets for fine-grained bird species detection (Reed etal, CVPR 16), visual question answering or activity recognition (Park etal CVPR 18) or self-driving explanations (Kim etal CVPR 18) exists. These datasets are natural test beds for evaluating natural language based explanations that does not contain the class label in the explanation and also a large portion of those images can not be labeled by an AMT annotator as the label of the object is not known to a lay person.

Correctness: As mentioned above, there is a confusion about the task the paper is addressing. Detailed feedback is provided in 'Weaknesses' and in 'Additional Feedback'.

Clarity: The paper is well written and easy to read. For the human experiments mentioned in the beginning of the second page, a visualisation of the annotation screen would help the reader, currently the section is quite descriptive and long. There are only six sentences (descriptions) are provided in the Figure 3. More examples would help the reader assess the diversity of the sentences.

Relation to Prior Work: Several works (as mentioned above) related to natural language explanations is missing.

Reproducibility: Yes

Additional Feedback: In addition to the questions raised in the weaknesses section, the reviewer would like some clarification on the issues below: 1. Line 130: How familiar the human annotator h is to the objects in the dataset? In other words, how many of the classes are easily recognisable and nameable by adult humans? 2. Line 134: As the annotator's is presented with two images, one of them belongs to the class of interest, how do the authors sample these pairs? In other words, if all the possible pairwise comparisons are not presented to the annotator (which may be infeasible given the size of the dataset), this data annotation paradigm may indicate some bias in the data. Have the authors measured the bias in the data? 3. Line 155: For the assessment task, the annotators are shown a short description of the class (arguably this is not a description of a class but rather a description of the image as the class name appears in the 'description'). Where do these descriptions come from? 4. Line 191: \psi(.) is indicated to be the output of a captioning model or an off-the-shelf sentence encoder that captures the semantic and textual similarities. Which framework has been chosen in the experimental evaluation to encode the sentences? 5. Line 199: If the descriptions contain enough information about the class, another evaluation metric could be classification accuracy. What is the top-1 accuracy of the automatic descriptions in ImageNet? 6. In addition to natural language, several works have proposed human annotated attributes as class-level descriptions. As attribute based learning seems to be quite relevant for this work, it would be nice to see results on that. Among the above mentioned datasets, CUB contains both human annotated attributes and sentence descriptions and a re-evaluation of the metric on this dataset would significantly improve the exposition of the paper.


Review 4

Summary and Contributions: The authors propose a method for evaluating the interpretability of visual concepts discovered by self-supervised learning techniques. To do so, they introduce two concepts, “visual learnability” and “describability”, which use humans to quantify the interpretability of arbitrary image groups. The former measures if a group is interpretable and coherent by assessing if humans can perform a small generalization test using a group of representative group images. The latter uses a similar test to measure how well a group can be summarized using a single natural language sentence instead. This summarization is done using an automatic class-level captioning algorithm. The authors collect captions for images using a pre-trained captioning model and, for each group, pick a representative sentence (based on intra-group proximity and inter-group distance). The authors conduct several experiments to evaluate the proposed metrics, the automatic class-level captioning, and the interpretability of the representations learned by two self-supervised learning approaches (SeLa and MoCo). They compare the learned groups with ImageNet and find that there exist self-discovered classes that are “interpretable” yet do not align naturally with ImageNet classes. They also conclude that the automatic class-level captioning method performs well to describe visual clusters.

Strengths: The paper attempts to provide mechanisms to interpret the representations learned by self-supervised learning techniques, which could be useful to determine their usefulness in downstream tasks without actually testing them on the said tasks. The automatic captioning system is also scalable, which can be used to provide temporary “interpretability labels” for self-supervised learning.

Weaknesses: Evaluations with human subjects are not scalable, so it can be cumbersome to apply the “visual learnability” (Semantic Coherence) metric to new self-supervised representation learning techniques. The automatic class-level captioning algorithm is currently limited and the experiments could be expanded (I detail the reasons why in the additional comments).

Correctness: The Semantic Coherence metric and its human evaluation seems appropriate. The Describability test could be expanded (I provide suggestions in the additional comments).

Clarity: The paper is well-written and easy to follow.

Relation to Prior Work: Yes.

Reproducibility: Yes

Additional Feedback: Some typos: - Line 64: “lead” -> “led” - Line 161: “X_{c}” -> “\hat{X}_{c}” - Line 183: Repeated sentence. - Line 212: “[w]e use data [from?] the training set of ImageNet” - Line 254: “clsutering” -> “clustering” - Line 270: “use” -> “using” - Line 272: “used [as?] class description” I am not certain that the experiments for Describability show that the automatically-generated captions are good at describing the class. It could be the case that the caption is not good and humans are doing the job of “abstracting” the content of the caption to generalize and correctly classify samples. For instance, a caption may simply contain one relevant word that links broadly with the concepts of one of the query images (e.g., caption mentions food, but it is the wrong food) and the other query image has no similar concepts (e.g., image of a car). Additionally, this evaluation depends heavily on the test samples, and no experiments are conducted to account for that (e.g., exploring hard negatives). I suggest creating another evaluation asking participants to asses whether an automatically-generated caption does a good job for representing a group of images (i.e., present a subset of images from a group and its corresponding caption and ask if the user thinks the caption represents the group, using a Likert scale to collect answers). Describing a class can usually require “going up a level” in abstractness. In my opinion, this is not going to be well captured by the proposed caption generator, which selects a position in the embedded space based on inter- and intra-group proximities. This leaves room for further work, in case the authors want to pursue that later on. After reading the authors rebuttal, I have increased my overall score of this submission.

[Author Response · NeurIPS 2020]

We thank the reviewers for their valuable feedback. All reviewers agree that the problem of evaluating unsupervised
representations/groupings is of high relevance and interest to the representation learning community. However, R3
might have misunderstood the goal/setup of the approach, which we hope to clarify further in this rebuttal.

**Reviewer 1.**    1. Why SeLa and MoCo?: They are representative of two important classes of unsupervised representation
learning algorithms: (1) deep clustering (SeLa) and (2) contrastive learning (MoCo). 2. Qualitative results (Fig. 3) &
failure cases: The results in Fig. 3 are average-case results. Additional results shown and discussed in the Appendix
also include failure cases (*e.g.* Fig. A1-h,j,l). In general, lower-purity clusters are less semantically coherent, thus the
class-level description is moe likely to fail. We will discuss this in the main paper. 3. Purity (L241): Entropy and
purity are measures adopted from cluster analysis. They are also briefly discussed in [3,8]. A cluster is highly pure if
the vast majority of its samples share a common label. Since in this case this label is manually provided, high purity
strongly correlates with high interpretability. 4. Error bars (Fig. 2): 95% CI, estimated per cluster.

**Reviewer 2.**    1. Soundness: choice of negative samples: Different functions for sampling
negatives correspond to exploring different aspects of the learned classes. In the paper we
considered the simplest choice (random negatives), which explores the overall learnability
of a class. We have since also considered hard negatives, *i.e.* sampling negatives only from
the class "most similar" to the target one, based on class centroids in feature space (SeLa
results in Tab. 1). This tests whether there are sufficient *fine-grained* differences between

Table 1: Semantic coherence with hard negatives.

| $\Pi$ | Random | Hard |
|---|---|---|
| $(0.3, 0.4]$ | 71.8 | 55.3 |
| $(0.4, 0.5]$ | 94.2 | 60.0 |
| $(0.5, 0.6]$ | 97.2 | 71.8 |
| $(0.6, 0.7]$ | 99.7 | 63.2 |
| $(0.7, 0.8]$ | 98.0 | 65.3 |
| $(0.8, 0.9]$ | 99.8 | 63.8 |
| $(0.9, 1.0]$ | 98.8 | 72.2 |

classes to be learnable by humans. The outcome of this experiment suggests that this is often
*not* the case, which is unsurprising given that the algorithms find a large number of clusters
(3000) and thus likely over-fragment the data. It also indicates that, while clusters are often
semantically coherent, they are not necessarily "complete", in the sense of encompassing *all* the images that should be
grouped together — finding the right number of clusters remains an open challenge in literature, and this experiments
emphasizes that. We will add these comments and results to the paper. 2. Significance/Usefulness/Scalability: The
goal of most studies in *interpretability* is to analyse a model independently of downstream tasks. For that, the use of
manual assessment is widespread despite its cost. Our contribution is of significance because it removes subjectivity in
this popular category of assessment methods [5,24,72]. The method we propose acts complementarily to downstream
tasks (*e.g.* training linear probes to compare against a pre-labelled dataset). Our findings are significant: we show
that these fixed labels do not necessarily align with the ones discovered automatically, *i.e.* less pure clusters are also
human-interpretable. As an example, in Fig. A1-g (Appendix), SeLa discovers a "newborn" class in ImageNet, which is
not part of the existing label set. 3. Cluster size: We agree; SeLa returns clusters of approximately the same size (min:
418, max: 435 samples), MoCo's $k$-means clusters vary in size (min: 1, max: 1238, median: 407). For fair comparisons,
we selected MoCo clusters with a min. size of 200 samples (mean: 465). 4. Desc. length and coherence: We observed
no correlation between sentence length and coherence. However, human-written descriptions tend to be short for pure
clusters since they can be easily described as a single concept (Pearson's $r = -0.38, p = 0.001$ between length/*purity*).

**Reviewer 3.**    1. Generating Visual Explanations; Reed/Park/Kim: There seems to be a misunderstanding about the
setting of our paper. The work and datasets on generating visual explanations address a very different problem, namely
to generate a description justifying individual image predictions. Instead: (1) we answer the question of whether a *given*
image grouping (unsupervised or not) is interpretable (can be learned by a human) and describable (can be captured by
a description; L26-35); (2) we consider descriptions for image *groups*, not individual images (L57-58); (3) our primary
goal is not to generate justifications, but to provide a human-based assessment method of interpretability in unsupervised
algorithms, instead of matching their output to pre-scripted labels (see R2.2). 2. Class label included in the description:
We use the term "class" to refer to an arbitrary image grouping, not a pre-specified category (L28-31). Similarly, by
"class-level description" we refer to the group caption; it is *not* a description of a known label and does not reveal a
"solution" — this is not applicable in our setting. 3. Few qualitative examples (descriptions): We have provided 4 pages
of qualitative examples and an interactive demo with image- and class-level captions for *all* clusters in the sup. mat.
4. How are pairs sampled?: Positives are sampled uniformly at random from the cluster in question; the negative is
sampled uniformly among all other clusters (L127-129). We include here experiments with hard negatives (see R2.1).
5. Where descriptions come from: They are either human-written or automatic (see Sec. 4, L204-207, 214-217). We
obtain one description per cluster in each case. 6. Sentence encoder: We use Sentence-BERT (L269-271).

**Reviewer 4.**    1. Scalability: Please see R2.2. 2. Exploring hard negatives: We have included
experiments with hard negatives (see R2.1). 3. Asking participants to assess captions: Note
that our method is designed precisely to avoid asking annotators to express a judgment (L36-40)
to remove subjectivity from the evaluation. Nevertheless, following your suggestion, we also
conducted an experiment asking participants to rate how well the auto-generated caption matches
an image group as a whole (scale: 1-worst to 5-best). The results (Fig. 1) follow the same trend
as the results presented in the paper. 4. "Going up a level" in abstractness: This is indeed a very
interesting direction that we have already started investigating as part of our future work.

Figure 1: Histogram of user ratings.

[Meta-Review · NeurIPS 2020]

The reviewers generally agreed that this paper provides a fresh perspective on a timely problem, i.e., evaluating the quality of learned representations via learnability and describability. After the rebuttal, all the reviewers (except R3) recommended Accept. R3 initially raised some concerns and the authors properly addressed them in the rebuttal. I concur with the reviewers' recommendation to accept the paper.